# Combined Effect of High Hydrostatic Pressure and Proteolytic Fraction P1G10 from *Vasconcellea cundinamarcensis* Latex against *Botrytis cinerea* in Grape Juice

**DOI:** 10.3390/foods12183400

**Published:** 2023-09-12

**Authors:** María José Torres-Ossandón, Luis Castillo, Elsa Uribe, Cristina Bilbao-Sainz, Kong Shun Ah-Hen, Antonio Vega-Gálvez

**Affiliations:** 1Laboratorio de Biotecnología y Microbiología Aplicada, Departamento en Ciencia y Tecnología de los Alimentos, Facultad Tecnológica, Universidad de Santiago de Chile, Alameda 3363, Estación Central, Santiago 9170022, Chile; 2Laboratorio de Bioquímica y Biología Molecular, Departamento de Biología, Universidad de La Serena, Avda. Raúl Bitrán 1305, La Serena 1700000, Chile; 3Departamento de Ingeniería en Alimentos, Universidad de La Serena, Avda. Raúl Bitrán 1305, La Serena 1700000, Chile; 4Instituto de Investigación Multidisciplinario en Ciencia y Tecnología, Universidad de La Serena, Avda. Raúl Bitrán 1305, La Serena 1700000, Chile; 5Healthy Processed Foods Research, U.S. Department of Agriculture, Albany, CA 94710, USA; 6Instituto de Ciencia y Tecnología de los Alimentos, Universidad Austral de Chile, Avda. Julio Sarrazín sn, Valdivia 5090000, Chile

**Keywords:** high-pressure processing, growth inhibition of *Botrytis*, papaya antimicrobial peptide, food safety, bioactives in grape juice

## Abstract

The effect of high hydrostatic pressure (HHP) and the proteolytic fraction P1G10 from papaya latex was studied to find out whether a synergy exists in the growth inhibition of *Botrytis cinerea* in grape juice, contributing to the improvement of conservation techniques and extending the shelf life and quality of food products. Grape juice (GJ) diluted to 16 °Brix with a water activity (a_w_) of 0.980 was prepared from a concentrated GJ and used in this study. Results indicated a 92% growth inhibition of *B. cinerea* when exposed to 1 mg/mL of P1G10 and 250 MPa/4 min of pressure treatment. The proximate composition and antioxidant compounds present in the GJ were not significantly affected after the treatments. Eight phenolic compounds and two flavonoids in GJ were identified and quantified, with values fluctuating between 12.77 ± 0.51 and 240.40 ± 20.9 mg/L in the control sample (0.1 MPa). The phenolic compounds showed a significant decrease after the applied treatments, with the HHP sample having a content of 65.4 ± 6.9 mg GAE/100 mL GJ. In conclusion, a synergistic effect at moderate HHP of 250 MPa/4 min with the addition of P1G10 was observed, and the successful development of a stable and acceptable GJ product was possible.

## 1. Introduction

The food industry is constantly seeking ways to improve preservation techniques to extend the shelf life and quality of food products. Thermal preservation methods are the most widespread due to the easy application of heat. However, high temperatures may be detrimental to nutritional qualities [1], leading to the development of non-thermal methods. Today, high hydrostatic pressure (HHP) has established itself as a prominent commercial method for food preservation. In this method, foods are pasteurized through the application of a high level of isostatic pressure from 100 to 1000 MPa to inactivate microorganisms [2,3,4,5]. However, the high investment costs of HHP equipment result in higher product prices compared to those processed by non-HHP or traditional pasteurization methods [6]. Hence, in recent years, studies on the development and application of HHP have been primarily focused on attributes such as health benefits and quality, which has led to HHP processing and the development of a panoply of healthy food products. HHP has also been used to enhance functional components and reduce the content of additives in food products [3]. An example is the preservation of fresh juices by HHP, which has achieved broad commercial success in the United States, United Kingdom, and Australia [2].

The inactivation of microorganisms by HHP depends on numerous factors, including the pressure level and the holding time. Generally, pressures ranging from 250 MPa to 500 MPa, along with pressurization times spanning 30 s to 20 min, have been used to control the growth of fungi and yeasts [7,8,9,10,11]. Moreover, the physicochemical characteristics of the food products, such as pH, water activity (a_w_), and acidity [12], must be considered when determining the conditions of pressurization. HHP is a batch process, and higher throughput can be achieved by decreasing processing time. However, this might increase the risk of inadequate pasteurization, so the use of combined treatments to produce synergy is looked at as an alternative process. Generally, this is achieved by following the principles of hurdle technology, where two or more antimicrobial agents can create a synergistic effect and increase the inactivation of microorganisms [13,14]. Several research studies have shown that HHP treatment combined with natural antimicrobials can achieve a synergistic effect and improve antimicrobial properties [14,15].

Currently, there are many antimicrobials derived from natural sources, most of which are secondary metabolites of plants [16,17], such as essential oils, phenolic compounds, flavonoids, and alkaloids [18]. Latex, which is a white, milky fluid found within the laticifer cells of unripe fruits [19], contains bioactive molecules with antifungal properties [20]. Latex contains antimicrobial peptides that can affect a broad spectrum of microorganisms (bacteria, fungi, viruses, and tumor cells) and are generally non-toxic and meet food safety requirements [21]. One peptide fraction, P1G10, from papaya (*Vasconcellea cundinamarcencis*) latex has shown important pharmacological properties [22,23] and considerable antifungal activity. P1G10 was found to inhibit spore germination and mycelium growth of *Colletotrichum gloeosporioides*, *Fusarium solani*, *Rhizoctonia solani*, *Neurospora* sp., and *Aspergillus niger* [24,25]. Meanwhile, a recent study examining the effect of P1G10 on *Botrytis cinerea* found that this fraction of the papaya latex inhibited mycelial growth, disturbed cell wall and cell membrane integrity, and altered the adhesion capacity of the fungus [26]. To the best of our knowledge, the effects of peptide fraction P1G10 during HHP treatment have not been investigated. P1G10 can be obtained by chromatographic separation on Sephadex-G10 and ultrafiltration. The bulk of the proteolytic fraction of P1G10 in *V. cundinamarcensis* is rich in cysteine proteinases, with a proteolytic activity about five times higher than that of papain from *Carica papaya* [27]. Altogether, 14 isoforms of cysteine proteinases have been identified in P1G10 from *V. cundinamarcensis* [19].

*B. cinerea* is a highly infectious pathogenic fungus that can cause significant economic losses in pre- and post-harvest fruits and vegetables [28]. It is part of the normal microflora found on grapes [29,30] and causes discoloration, an unpleasant smell, and difficulties during the filtration of grape musts. Heat treatments are usually employed to avoid the presence of *B. cinerea* in grape juices. However, the application of heat could affect the phenolic composition of grape juices. Consequently, alternative treatments are required to maintain the quality of grape juices and their bioactive properties [14]. Related research on the antimicrobial effects of HHP treatment or combined HHP treatment with natural antimicrobials on the growth of *B. cinerea* reported that 400 MPa/5 min without preservatives and with avocado seed acetogenins (661 μg/mL) presented viable conidia in strawberry puree. The use of a higher concentration of avocado seed acetogenins (7500 μg/mL) combined with the same pressure treatment (400 MPa/5 min) achieved the inactivation of *B. cinerea* conidia. In addition, the authors showed that the oxidative stress induced by the fungus produced significant losses of phytonutrients [31].

Therefore, in this study, the antimicrobial effects of HHP processing and a combined HHP treatment with the addition of P1G10 (HHP + P1G10) on the growth of *B. cinerea* were assessed. Additionally, we evaluated the effects of the treatments on physicochemical parameters, bioactive compounds, antioxidant potential, and various quality parameters of a white grape (*Vitis vinifera*) juice.

## 2. Materials and Methods

### 2.1. Samples

Grape juice (GJ) from variety Pedro Jiménez with a solid content of 16 °Brix was prepared from unpasteurized grape juice concentrate (GJC), elaborated by the company Cooperativa Capel, Viña Francisco de Aguirre S.A., Ovalle, Coquimbo, Chile. Prior to HHP assays (HHP and combined HHP + P1G10 treatments), microbiological analysis of the GJ sample was performed and showed no mold growth. P1G10, a proteolytic fraction rich in cysteine proteinases, was obtained from the latex of *V. cundinamarcensis* as described in previous work by Torres-Ossandón et al. [26].

### 2.2. High Hydrostatic Pressure (HHP) and Combined HHP + P1G10 Treatments

The GJ samples were inoculated with *B. cinerea* to an initial count of 1 × 10^4^ conidia/mL before HHP and combined HHP + P1G10 treatments. The samples were packed in flexible polyethylene bags and processed in HHP equipment (2 L cylindrical container, Avure Technologies Incorporated, Kent, WA, USA), where water was the pressure-transmitting medium. Pressure levels of 100, 200, and 250 MPa for 2 and 4 min were applied to the samples at room temperature (20 ± 1 °C). The control sample was an unpressurized sample at atmospheric pressure (0.1 MPa) and room temperature. For the combined treatment, P1G10 at a concentration of 1 mg/mL, based on results from previous work [26], was added to the inoculated samples to examine their effects on *B. cinerea* [26]. The control was a pressurized sample without P1G10. After each treatment, the samples were incubated at 22 ± 1 °C for 1 h before microbiological analysis.

### 2.3. Microbiological Analysis

The analysis of *B. cinerea* in the GJ samples (control and treated samples) was performed according to Reyes et al. [32]. Decimal dilutions with 0.1% peptone solution were performed, and duplicates of at least three dilutions were plated on the corresponding culture media. To assess the mold growth, 1.0 mL of the initial (10^−1^) dilution was spread-plated on three different plates (0.1, 0.3, and 0.4 mL) of Dichloran Rose Bengal Chloramphenicol (DRBC) agar (Difco, Detroit, MI, USA). Then, 0.1 mL of each subsequent dilution was spread on individual DRBC plates. The inoculated plates were then incubated for 5 days at 22 ± 1 °C, and plates with 15–300 colonies were counted.

### 2.4. Proximate Composition and Physicochemical Analyses

The proximate composition was determined using the methods of the Association of Official Analytical Chemists (AOAC 1990) [33]. This included moisture content (Method N° 934.06), total crude protein (Method N° 960.52), lipid content (Method N° 960.39), and crude ash (Method N° 923.03). The total carbohydrate content was estimated by the difference. Physicochemical characteristics of the samples were measured, including pH with a pH-meter (Orion Star A320, Thermo Scientific Inc., Fort Collins, CO, USA), titratable acidity expressed as gram tartaric acid per 100 mL of GJ, soluble solid content (°Brix) measured with an Abbé refractometer (DR-A1, Atago, Tokyo, Japan), and the surface color measured with a colorimeter (HunterLab, MiniScanTM XE Plus, Reston, VA, USA). The total color difference (Δ*E*) was calculated using Equation (1).
(1)$$∆E=\sqrt{\left[{\left(L*-L_{0}*\right)}^{2}+{\left(a*-a_{0}*\right)}^{2}+{\left(b*-b_{0}*\right)}^{2}\right]}$$
where *L** is the lightness of the sample, *L*_0_* is the lightness of the fresh sample, *a** is the redness of the sample, *a*_0_* is the redness of the fresh sample, *b** is the yellowness of the sample, and *b*_0_* is the yellowness of the fresh sample.

### 2.5. Total Flavonoid Content

Total flavonoid content (TFC) was determined by a colorimetric assay as described by Zhishen et al. [34]. First, 0.5 mL of GJ was placed in a 5 mL tube containing 2 mL of distilled water. At time zero, 0.15 mL of an aqueous solution of NaNO_2_ (5 g/100 mL) was added to the tube. After 5 min, 0.15 mL of aqueous AlCl_3_ solution (10 g/100 mL) was added to the mixture. At 6 min, 1 mL of 1 M NaOH was added to the mixture, followed by immediate dilution with 1.2 mL of distilled water and thorough mixing. The absorbance of the samples was read at 415 nm against a blank containing water. A calibration curve was prepared using 25 to 300 μg quercetin/mL. Results were expressed as mg quercetin equivalents/100 mL GJ (mg QE/100 mL GJ).

### 2.6. Phenolic Compounds Extraction and Determination

Phenolic compounds from GJ were extracted using a conventional discontinuous method [35]. One hundred mL of GJ was extracted four times with ethyl ether (1:1). After evaporation to dryness at 37 °C using a rotary evaporator (IKA RV 10, Staufen, Germany), 1 mL of methanol–formic acid (99:1) was added to dissolve the residue, which that was then filtered through a 0.45 µm membrane filter (Merck Millipore KGaA, Darmstadt, Germany).

Total phenolic content (TPC) was determined using the Folin-Ciocalteu method [36]. TPC was calculated as mg gallic acid equivalent/100 mL GJ (mg GAE/100 mL GJ) using a calibration curve created using gallic acid standard solutions.

The analysis of the individual phenolic compounds was carried out using HPLC (Agilent 1200, equipped with a high-pressure pump, automatic injector, and a diode array detector (DAD) system controlled by ChemStation software (V04.02.096) with a Kromasil 100-5C18 analytical column (250 × 4.6 mm; 5 μm particle size), Eka Chemical, Sweden). The injection volume was 10 µL, the flow rate at 25 °C was 0.7 mL/min, and the eluates were monitored at 280, 310, and 370 nm. The mobile phase consisted of two solvents: solvent A (formic acid 0.3%, pH 2.6) and solvent B (100% acetonitrile). The elution was as follows: 0 to 16 min with 87% A and 13% B; 16 to 23 min with 45% A and 55% B; 23 to 25 min with 40% A and 60% B; 25 to 30 min with 87% A and 13% B; and back to initial conditions within 4 min. The results were expressed as mg/mL GJ. All measurements were performed in triplicate.

### 2.7. Antioxidant Potential

The antioxidant potential of GJ was determined by the DPPH (2,2 diphenyl-1-picrylhydrazyl) and ORAC (Oxygen Radical Absorbance Capacity) assays. The DPPH assay is based on the evaluation of the free radical scavenging capacity, according to Brand-Williams et al. [37]. One hundred μL of GJ and 3.9 mL of 50 μM DPPH in 80% (*v*/*v*) ethanol were vortex-mixed for 30 s and left in the dark at room temperature for 30 min. The absorbance was read at 517 nm using a UV–VIS spectrophotometer (Orion Aqua Mate 8000, Thermo Scientific, Madison, WI, USA). A blank sample was prepared without the addition of GJ.

The ORAC assay was performed using a Multilabel Plate Reader (Perkin Elmer Victor X3, Turku, Finland) as described by Torres-Ossandón et al. [38]. A total of 40 μL of GJ diluted with water at a ratio of 100:1 was prepared in phosphate buffer (pH 7.0). The antioxidant capacity was measured using an excitation wavelength of 485 nm and an emission wavelength of 520 nm. The Trolox standard was used for both methods to construct calibration curves, and the results were expressed as µmol Trolox Equivalent per 100 mL GJ (µmol TE/100 mL GJ).

### 2.8. Sensory Evaluation

A semi-trained panel of 28 tasters carried out the sensory evaluation of the grape juice using a preference test. Two coded samples of the grape juice were presented to the panelists. The samples were subjected to HHP treatments at 250 MPa/4 min and 250 MPa/4 min with added P1G10. The judges were asked to evaluate the attributes of acidity, sweetness, color, and odor of the samples. In addition, they were consulted for their preferences according to a hedonic scale. The hedonic scales ranged from 1 (Like Extremely) to 9 (Dislike Extremely). Samples were served at room temperature in a transparent glass cup coded with three-digit random numbers. In this way, preferences were determined according to the different attributes [39].

### 2.9. Statistical Analysis

All the treatments were repeated at least three times, and each treatment was conducted with at least three duplicates. A two-way analysis of variance (ANOVA) (Statgraphics Plus^®^ 5.1 software, Statistical Graphics Corp., Herndon, VA, USA) was used to demonstrate significant differences among samples. Significance testing was performed using Fisher’s least significant difference (LSD) test. Differences were accepted at confidence levels of 95% and 99%. The Multiple Range Test (MRT) was performed to assess the presence of homogeneous groups within each of the analyzed parameters.

## 3. Results

### 3.1. Inactivation of Conidia of B. cinerea by HHP and HHP-P1G10 Treatments

In this study, grape juice with a soluble solid content equivalent to 16 °Brix, which is the recommended level for reconstituted fruit juices according to CODEX STAN 247-2005 [40], was used for the assays.

In Figure 1, plate count and survival curves of *B. cinerea* after different HHP and combined HHP + P1G10 treatments are depicted. It can be seen that higher pressures and longer treatment times led to greater inactivation of the *B. cinerea* conidia and, consequently, growth inhibition of the fungus. A significant difference in survival occurred between the HHP treatments with and without P1G10. For the HHP treatment at 250 MPa/4 min, a survival rate of 22% of the fungi was determined, while after the combined treatment (HHP + P1G10), only 8% survival of the fungi was observed. Therefore, the combined treatment (HHP + P1G10) at 250 MPa/4 min resulted in a *B. cinerea* count of about three times less than that without P1G10. The survival of *B. cinerea* determined at different concentrations of P1G10 showed that 1 mg/mL would inhibit 50% of mycelium growth after 72 h incubation [26]. This suggests the inhibitory effect of the combined treatments is greater than the effects of each treatment applied separately [41], which may be considered a “synergistic effect.” Thus, treatments with synergistic effects allow a combination of HHP treatments at lower pressures and the addition of natural antimicrobials at lower concentrations to reduce the microbial load with less impact on the quality of the foods [42].

It has been reported that a reduction in the pH of the medium causes a progressive increase in cellular sensitivity to pressure [43]. In this study, the average pH of grape juice was maintained at 4.00 ± 0.02, which was the optimal growth pH for *B. cinerea*. Therefore, the pH was not a deterrent factor in the combined HHP + P1G10 treatment against *B. cinerea* in the grape juice.

The results of this study showed that the conidia of *B. cinerea* that survived the HHP treatment were potentially damaged and inactivated by P1G10 since the applied pressures in the combined treatments did not affect the antifungal activity of P1G10 (Appendix A: Proteolytic activity of P1G10 (1 mg/mL) at different pressure/time conditions).

### 3.2. Determination of Physicochemical Parameters

The proximate compositions of the grape juice samples are presented in Table 1. The results are in a similar range to those reported in the literature [44]. In general, no significant changes occurred during any of the grape juice treatments. However, lipid content in the presence of P1G10 with or without HHP treatment was observed to be higher but statistically not significantly different at a confidence level of 95%. The other physicochemical properties, such as acidity, soluble solid content, and water activity, did not show significant differences in the treated grape juice compared to the control sample (0.1 MPa), although pH increased slightly in the samples pressurized at 250 MPa. On the other hand, the changes in chromatic parameters were significant (*p* < 0.05) in the treated juices. Lightness (*L** = 0 yields black, and *L** = 100 indicates diffuse white) increased significantly (*p* < 0.05), although slightly in P1G10 and HHP (250 MPa/4 min) samples. Conversely, in the combined HHP + P1G10 treated sample, a slight but significant (*p* < 0.05) decrease in lightness was observed. The *a** value (negative values indicate green and positive values indicate red) showed a tendency to increase in a negative direction in the presence of P1G10, while with the HHP-treated sample, an increase in the positive value occurred. In the combined treatment (HHP + P1G10), as expected, an attenuated change in the negative direction was observed. On the other hand, the *b** value (negative values indicate blue, and positive values indicate yellow) of the treated grape juice samples showed no significant (*p* < 0.05) differences from the control (Table 1). Overall, all treated samples showed significant (*p* < 0.05) color differences compared to the control, although the HHP + P1G10 sample showed the greatest color difference with respect to the absolute Δ*E* value. The observed changes in color values of the HHP-treated samples are consistent with results from reported studies [10]. The addition of P1G10 to grape juice seemed to cause a significant (*p* < 0.05) increase in the green component of the juice.

### 3.3. Antioxidant Potential and Phenolic Profile

The antioxidant potential, as determined by ORAC and DPPH assays, as well as the total flavonoid and total polyphenol contents of the grape juice samples, are shown in Figure 2. The control sample (0.1 MPa) had an antioxidant potential of 2450.8 ± 90.8 µmol TE/100 mL GJ as determined by the ORAC assays (Figure 2a), while the DPPH assays gave a value of 117.4 ± 1.4 µmol TE/100 mL GJ (Figure 2b). The results of the ORAC and DPPH assays differ greatly since both methods are based on different oxidation reaction mechanisms [45]. Nonetheless, both methods showed that the treatments of the grape juice samples, either by addition of P1G10 or by pressurizing with or without P1G10, did not severely alter the antioxidant potential. The addition of P1G10 caused a significant (*p* < 0.05) decrease in the ORAC value, down to 1790.8 ± 152.3 µmol TE/100 mL GJ (Figure 2a). In contrast, the DPPH values increased slightly but significantly in all treatments, with the P1G10 sample reaching a value of 134.8 ± 1.4 µmol TE/100 mL GJ (Figure 2b), which was slightly higher than that of the pressurized samples. It has been reported that HHP treatment of cranberry juice using pressures of 200 to 400 MPa resulted in similar or higher total antioxidant capacity when compared to fresh juice [46].

The unpressurized juice (control) has a total flavonoid content (TFC) of 19.9 mg QE/100 mL GJ (Figure 2c) and a total polyphenol content of 100.3 mg GAE/100 mL GJ (Figure 2d). In all the treated samples, slight but significant (*p* < 0.05) increases in TFC were determined, being highest in the P1G10 sample (24.1 ± 0.6 mg QE/100 mL GJ), which could be related to enzymatic activities [47]. On the other hand, total polyphenol content (TPC) showed a significant (*p* < 0.05) decrease in all treated samples, with the greatest loss in the HHP sample with a TPC of 65.4 ± 6.9 mg GAE/100 mL GJ.

Table 2 shows the phenolic profile of the grape juice samples after different treatments. Similar results have been reported for concentrated grape juice after HHP treatments [41]. In general, significant (*p* < 0.05) decreases in the content of the polyphenolic compounds occurred after the different treatments, with the HHP samples showing the greatest loss for all analyzed phenolics. On the other hand, the samples treated with P1G10 and also in combination with pressurization at 250 MPa/4 min showed a significant (*p* < 0.05) decrease only in the contents of some of the phenolic compounds, namely protocatechuic acid, p-coumaric acid, tyrosol, and ellagic acid. Nonetheless, the changes may be considered to be hardly relevant in terms of quality.

### 3.4. Sensory Evaluation of Grape Juice Treated by HHP and Combined HHP + P1G10

Figure 3 shows the results of the preference tests for the different sensory attributes of the two treated grape juice samples. The responses of the panelists are presented in percentages, considering the responses of all judges as 100%. The grape juice from the combined treatment of 250 MPa/4 min + P1G10 had higher preference scores for acidity, sweetness, and odor, whereas the sample from the HHP treatment of 250 MPa/4 min had a higher preference score for color (Figure 3a). From Table 1, it can be seen that the 250 MPa/4 min + P1G10 sample had a lower lightness value, which might have affected the preferences of the judges. High-pressure technology has been shown to produce juices with fresh and natural attributes as well as natural-looking colors, aspects that all consumers value today [48].

In addition to scoring sensory attributes, the panelists were consulted for general appreciation of the juice according to a hedonic scale. The results are presented in Figure 3b. It shows that ten judges favored the decision “like very much” for the juice from the combined treatment HHP + P1G10 compared to eight judges who preferred the juice from only the HHP treatment, and nine judges decided “like moderately” for both juices. Two panelists opted for “dislike very much” and another one for “dislike slightly” for the sample from the combined treatment, so it may be possible that the presence of P1G10 may have been perceived. Nonetheless, acceptance of the juices predominated in both cases, so it may be concluded that the overall HHP treatment would not affect gustatory perception significantly. Studies of sensory evaluation in products subjected to HHP treatments are still scarce. Polydera, Stoforos, and Taoukis [49] and Laboissière, Deliza, and Barros-Marcellini [50] conducted similar studies regarding the effects of HHP on sensory attributes in tomato juice, passion fruit juice, guava puree, and pineapple juice.

## 4. Discussion

Previous studies have shown that vegetative bacteria, yeasts, and molds can be inactivated by HHP treatments above 250 MPa [51]. Also, the structure of foods, such as the mechanical distribution of water, the chemical distribution of food preservatives, and physical constraints on mobility, affect the proliferation of microorganisms and, consequently, the foods’ shelf life [52]. These factors can enhance or reduce the efficiency of HHP treatments. The effects of HHP on the growth of *B. cinerea* in grape juice were shown to depend on the soluble solid content [53]. In this study, the results suggested that the combined treatment of GJ (16 °Brix) by HHP at 250 MPa/4 min in the presence of the peptide fraction P1G10 at a concentration of 1 mg/mL GJ had an inhibitory effect on *B. cinerea* greater than that of either HHP at 250 MPa/4 min or the addition of P1G10 applied separately. This implied a synergistic effect of the combined treatment, as reported in several studies on the synergistic effect of high-pressure technology and natural antimicrobials on microbial growth. Most of these studies on the effects of combined treatments are focused mainly on bacteria. Qi et al. [54] demonstrated that food systems with neutral pH treated at 500 MPa for 15 min with the addition of nisin (200 IU/mL) reduced the population of *Bacillus subtilis* and increased its membrane permeability by 10–60%. Somolinos et al. [55] also investigated the application of high pressure combined with citral and tert-butyl hydroquinone (TBHQ) to inactivate *Escherichia coli* and *Listeria monocytogenes* cells. TBHQ alone was not lethal to *E. coli* or *L. monocytogenes* but acted synergistically with HHP, resulting in a decrease in viability of >5 log10 cycles for both organisms. The combination of citral and HHP also showed a synergistic effect, allowing the achievement of either a higher degree of inactivation or a higher proportion of sub-lethally injured cells. HHP produced sublethal lesions in bacteria, with the membrane being a primary site of damage from pressure [43]. Therefore, cells with damaged membranes may show greater sensitivity to antimicrobials [56]. Zhao et al. [57] demonstrated a synergistic effect between 100 IU/mL of nisin combined with 400 MPa/4 min and 500 MPa/2 min in the inactivation of total aerobic bacteria in cucumber juice. In addition, these authors evaluated the synergistic treatment of fungi and yeasts and observed that the growth levels of fungi and yeasts were below the detection limit.

These results coincide with different studies that showed the behavior of fungi and yeasts subjected to pressure treatments up to 300 MPa [5]. Based on these results, a moderate pressure level (pressures < 400 MPa) was applied in the assays. The sensitivity of *B. cinerea* to high pressure in the grape juice (16 °Brix) was found to increase with pressure level. However, a minimum pressure level of 250 MPa for 4 min should be achieved. This is comparable to the range of pressure between 200 and 300 MPa recommended for microbial inactivation by HHP in solid food, which is indeed more complex than in a water or buffer system. Furthermore, nutrient-rich solid media usually contain substances that can protect and allow the recovery of the microorganism against damage [56].

Torres-Ossandón et al. [26] also showed that P1G10 disturbed the plasmatic membrane and cell wall of *B. cinerea*. The fundamental structure of membranes is the phospholipid bilayer, which forms a stable barrier between two aqueous compartments. The membranes also contain proteins responsible for the selective transport of molecules and cell-to-cell recognition. Some membranes can contain up to 50% protein by weight [58]. Therefore, peptidases can degrade membrane proteins by altering their permeability and allowing the uncontrolled flow of substances through the cell [24]. Regarding the pharmacological properties that have been attributed to P1G10, a recent study that shows a protective effect on gastric ulcers and healing activities in different skin lesions in mice and humans stands out [59]. Therefore, in addition to the antifungal activity that P1G10 possesses, this fraction could add value to grape juice. Studies on additives or extracts of natural origin have increased; however, they are still underused, and their use could expand the limits in the development of new foods, impacting the safety and preservation of food quality due to the fact that they are considered relatively safer and pose no risk to consumers [60].

High-acid food products like fruit juice are prone to contamination by mold and yeast spore-formers. It has been reported that in black/red raspberry juice treated with HHP, the change in pH value showed dependency on the pressure and time of the treatment. In the investigated pressure range between 400 and 600 MPa for treatment times of 2 min and 10 min, the lowest pH was observed at the lower pressure of 400 MPa and shorter treatment time of 2 min [61]. However, pH fluctuated between 3.36 ± 0.01 and 3.45 ± 0.01, which indicated a low change in acidity. In this study, pH increased from 3.45 ± 0.01 in the untreated sample to 4.03 ± 0.01 in the HHP 250 MPa/2 min sample. In the combined treatment of HHP 250 MPa/4 min + P1G10, the pH was slightly lower (Table 1). Therefore, HHP treatment that may lead to the rupture of the intracellular matrix of cells did not contribute to an increase in soluble solids (Table 1) but rather to an attenuation of acidity, a mechanism that could require further investigation to elucidate.

The alterations in phenolic compounds in HHP-treated juice could be related to increased oxidative stress induced by enzymes such as peroxidase [62]. P1G10, on the other hand, did not seem to cause oxidative stress since the phenolic compounds in the juice samples of the combined HHP + P1G10 treatment were at a significantly higher level (Figure 2d and Table 2). Nonetheless, the impact on the phenolic profiles is low. As reported, it could be much less than its occurrence during thermal processing [62], which could be considered an advantage of HHP processing.

The increase or conservation of antioxidant compounds in grape juice after the addition of P1G10 could be attributed to possible molecules with antioxidant potential found in the P1G10 fraction mixture that was purified through a Sephadex-G10 column with an exclusion size limit of proteins of <700 Da (Sigma-Aldrich, Burlington, MA, USA). However, this affirmation might need further experimental validation. Different peptides from food sources with antioxidant potential have been widely studied, especially proteins derived from foods of animal origin and some vegetable proteins in hemp, rice, and peanuts [63,64]. Their bioactive properties were mainly related to enzyme specificity, degree of hydrolysis, amino acid composition, structure, and hydrophobicity [65].

As for the effect of the HHP processing with or without the addition of P1G10 on the sensorial properties of the GJ, it may be considered almost imperceptible, which has been reported in various studies [48,49].

## 5. Conclusions

The effect of P1G10 in combination with a pressurization treatment (HHP + P1G10) showed a greater inhibitory effect on *B. cinerea* than that of HHP or the addition of P1G10 applied separately. On the other hand, the effect of the peptide fraction P1G10 on antioxidant compounds and juice quality can be considered non-significant. The sensory properties of the juice also remained unaltered with the addition of P1G10, which was almost imperceptible except for a slight color change. In summary, P1G10 could be a useful antifungal additive for the prevention of the development of *B. cinerea* fungus in grape juices. Further studies with experimental validation should be conducted to explain the mechanism of action that underlies the detected synergistic effect of HHP treatment with P1G10.

## Figures and Tables

**Figure 1 foods-12-03400-f001:**
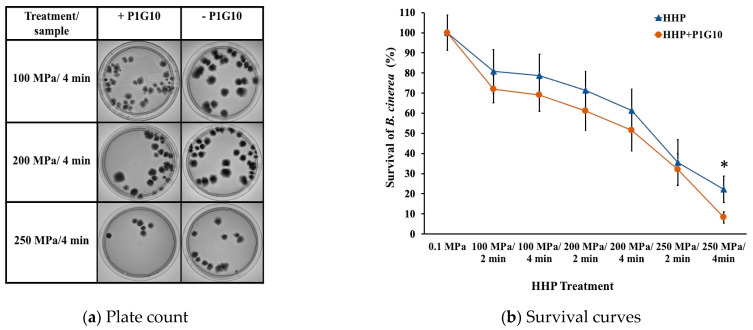
Plate count (**a**) and survival curves of *B. cinerea* (**b**) after HHP and combined HHP + P1G10 treatments. * Indicates significant differences (*p* < 0.05).

**Figure 2 foods-12-03400-f002:**
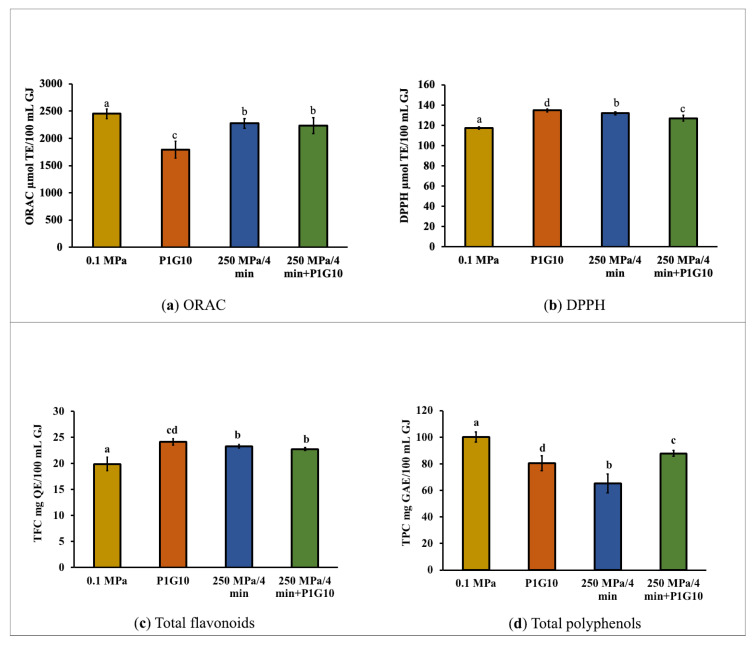
Antioxidant capacity (**a**) ORAC, (**b**) DPPH and contents of bioactives, (**c**) TFC, (**d**) TPC. Different letter indicates significant differences (*p* < 0.05).

**Figure 3 foods-12-03400-f003:**
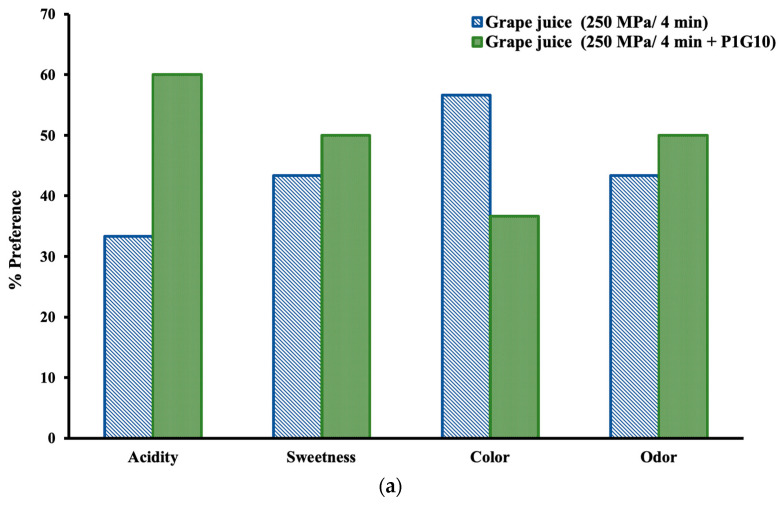
Sensory evaluation: (**a**) Preference test for different attributes; (**b**) Hedonic scale for appreciation.

**Table 1 foods-12-03400-t001:** Physicochemical characterization of grape juice after different treatments.

Physicochemical Analysis		Treatments		
g/100 mL	0.1 MPa	P1G10	250 MPa/ 4 min	250 MPa/ 4 min + P1G10
Moisture	80.56 ± 0.36 ^a^	80.42 ± 0.21 ^a^	80.80 ± 0.16 ^a^	80.78 ± 0.10 ^a^
Lipid	0.08 ± 0.05 ^ab^	0.20 ± 0.08 ^a^	0.05 ± 0.00 ^b^	0.20 ± 0.07 ^a^
Protein	0.46 ± 0.07 ^a^	0.47 ± 0.05 ^a^	0.46 ± 0.04 ^a^	0.46 ± 0.04 ^a^
Ash	0.15 ± 0.02^a^	0.13 ± 0.05 ^a^	0.16 ± 0.02 ^a^	0.14 ± 0.03 ^a^
Total carbohydrates	18.66 ± 0.36 ^a^	80.42 ± 0.21 ^a^	18.54 ± 0.10 ^a^	18.6 ± 0.55 ^a^
%Acidity	0.13 ± 0.01 ^a^	0.12 ± 0.01 ^a^	0.11 ± 0.01 ^a^	0.12 ± 0.01 ^a^
Soluble solid (°Brix)	15.6 ± 0.50 ^a^	15.5 ± 0.30 ^a^	15.8 ± 0.50 ^a^	15.3 ± 0.50 ^a^
pH	3.45 ± 0.01 ^c^	3.46 ± 0.01 ^c^	4.03 ± 0.01 ^a^	3.96 ± 0.01 ^b^
a_w_	0.980 ± 0.011 ^a^	0.987 ± 0.010 ^a^	0.976 ± 0.010 ^a^	0.987 ± 0.010 ^a^
Chromatic parameters				
*L**	64.50 ± 0.05 ^c^	65.56 ± 0.32 ^b^	65.88 ± 0.23 ^a^	62.52 ± 0.15 ^d^
*a**	−0.05 ± 0.03 ^b^	−0.68 ± 0.03 ^d^	0.97 ± 0.02 ^a^	−0.25 ± 0.02 ^c^
*b**	14.15 ± 0.21 ^a^	11.96 ± 0.76 ^a^	12.55 ± 5.30 ^a^	11.47 ± 0.12 ^a^
∆*E*	0	28.36 ± 0.80 ^a^	26.78 ± 4.39 ^a^	28.52 ± 1.69 ^a^

^a–d^ Different letter in the same row indicates significant differences (*p* < 0.05).

**Table 2 foods-12-03400-t002:** Profile of phenolic compounds in grape juice.

Phenolic Compounds		Treatments		
mg/L	0.1 MPa	P1G10	250 MPa/ 4 min	250 MPa/ 4 min + P1G10
Gallic acid	32.02 ± 2.83 ^a^	31.82 ± 2.77 ^a^	26.42 ± 1.56 ^b^	30.23 ± 2.39 ^a^
Protocatechuic acid	38.54 ± 3.03 ^a^	33.04 ± 3.11 ^b^	28.69 ± 1.87 ^b^	32.25 ± 2.83 ^b^
Vanillic acid	12.77 ± 0.51 ^a^	12.63 ± 0.94 ^a^	9.70 ± 0.23 ^b^	10.76 ± 0.95 ^b^
Caffeic acid	13.59 ± 0.56 ^a^	12.19 ± 1.43 ^ab^	10.93 ± 1.24 ^b^	12.38 ± 1.35 ^ab^
p-coumaric acid	15.14 ± 0.51 ^a^	13.46 ± 0.78 ^b^	11.54 ± 0.74 ^c^	12.20 ± 0.71 ^bc^
Tyrosol	37.51 ± 3.59 ^a^	34.84 ± 4.48 ^b^	29.82 ± 1.67 ^ab^	34.39 ± 3.39 ^ab^
Ferulic acid	12.93 ± 0.64 ^a^	11.55 ± 1.19 ^ab^	10.53 ± 0.67 ^b^	11.08 ± 0.63 ^b^
Ellagic acid	15.21 ± 1.08 ^a^	11.91 ± 1.60 ^b^	12.45 ± 1.17 ^b^	11.71 ± 1.02 ^b^
Catechin	240.40 ± 20.9 ^a^	212.51 ± 13.62 ^ab^	178.11 ± 14.70 ^b^	199.82 ± 10.65 ^ab^
Quercetin	37.90 ± 2.72 ^a^	33.70 ± 1.88 ^a^	28.44 ± 3.11 ^b^	26.50 ± 2.11 ^b^

^a–c^ Different letter in the same row indicates significant differences (*p* < 0.05).

## Data Availability

The data used to support the findings of this study can be made available by the corresponding author upon request.

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
