# Peer review of "Combined Effect of High Hydrostatic Pressure and Proteolytic Fraction P1G10 from *Vasconcellea cundinamarcensis* Latex against *Botrytis cinerea* in Grape Juice"

_foods, 2023, doi:10.3390/foods12183400_

Round 1
Reviewer 1 Report
Review report Foods-2585227
Combined effect of high hydrostatic pressure and proteolytic 2 fraction P1G10 from Vasconcellea cundinamarcensis latex 3 against Botrytis cinerea in grape juice.
The paper summarised the use of High hydrostatic pressure (HHP) and combined HHP+P1G10 treatments on latex obtained from Vasconcellea cundinamarcensis with the intention of finding the most acceptable stable and acceptable GJ product with little or no botrytis cinerea fungal strain as the main pathogenic fungus infecting the grapes. Prolonged shelf life is key in food stuffs as it guarantees great food security. Proximate composition, that includes moisture content, total crude protein and carbohydrate contents, lipid content, and crude ash, and other physicochemical analyses were used. Total flavonoids and phenolic compounds were also incorporated into the paper. The antioxidant activity of GJ was also determined in vitro. More importantly, sensory evaluation of the sample was undertaken. I recommend minor corrections are made.
Abstract
Line 27-28
Please remove the names of the phenolic acids identified. It gives a reader a need-to-know which phenolics were identified.
Line 29
Please remove the words “slight but”.
Introduction
To the best of my knowledge the introduction is well written. Unfortunately, I recommend the authors of the manuscript to reduce the length of the introduction without removing any of the references thereof. Please summarise it more.
Methodology
Line 161.
Which instrument was used to dry the samples? Dry air?
Results
Well presented. In my opinion, the references from 37 to 49 must be removed and be incorporated into the Discussions. Take an example of some of the lines that you need to remove. Line 263, its ideal to put numbers that fluctuated in short. Otherwise, one would ask, what would be the reason for the fluctuation of Lipids? Otherwise, results must be combined with the Discussion which is also recommended by the Journal.
Discussions
Well written. Congratulations to the authors.
References.
Well written. Congratulations to the authors.
Author Response
Reviewer 1
Comments and Suggestions for Authors
Review report Foods-2585227
Combined effect of high hydrostatic pressure and proteolytic 2 fraction P1G10 from Vasconcellea cundinamarcensis latex 3 against Botrytis cinerea in grape juice.
The paper summarised the use of High hydrostatic pressure (HHP) and combined HHP+P1G10 treatments on latex obtained from Vasconcellea cundinamarcensis with the intention of finding the most acceptable stable and acceptable GJ product with little or no botrytis cinerea fungal strain as the main pathogenic fungus infecting the grapes. Prolonged shelf life is key in food stuffs as it guarantees great food security. Proximate composition, that includes moisture content, total crude protein and carbohydrate contents, lipid content, and crude ash, and other physicochemical analyses were used. Total flavonoids and phenolic compounds were also incorporated into the paper. The antioxidant activity of GJ was also determined in vitro. More importantly, sensory evaluation of the sample was undertaken. I recommend minor corrections are made.
Authors: We thanks reviewer for appreciation of our work. We did our best to correct the manuscript as suggested.
Reviewer 1
Abstract
Line 27-28
Please remove the names of the phenolic acids identified. It gives a reader a need-to-know which phenolics were identified.
Authors: The names of the phenolics: “gallic acid, protocatechuic acid, vanillic acid, caffeic acid, p-coumaric acid, tyrosol, ferulic acid, ellagic acid, catechin and quercetin” were removed.
Reviewer 1
Line 29
Please remove the words “slight but”.
Authors: Corrected as suggested.
Reviewer 1
Introduction
To the best of my knowledge the introduction is well written. Unfortunately, I recommend the authors of the manuscript to reduce the length of the introduction without removing any of the references thereof. Please summarise it more.
Authors: We thanks reviewer for the suggestion. We have shortened some phrases and removed sections highlighted in yellow; however, we believe the introduction is concise and any reduction may not be appropriate.
Reviewer 1
Methodology
Line 161.
Which instrument was used to dry the samples? Dry air?
Authors: A rotary evaporator was used. The text was amended as required.
Reviewer 1
Results
Well presented. In my opinion, the references from 37 to 49 must be removed and be incorporated into the Discussions. Take an example of some of the lines that you need to remove. Line 263, its ideal to put numbers that fluctuated in short. Otherwise, one would ask, what would be the reason for the fluctuation of Lipids? Otherwise, results must be combined with the Discussion which is also recommended by the Journal.
Authors: The following text was removed and incorporated into discussion section as suggested: “Previous studies have shown that vegetative bacteria, yeasts, and molds can be inactivated by HHP treatments above 250 MPa [37]. Also, the structure of foods, such as mechanical distribution of water, chemical distribution of food preservatives, and physical constraints on mobility affect the proliferation of microorganisms and consequently the foods’ shelf-life [38], which can enhance or reduce the efficiency of HHP treatments. The effects of HHP on the growth of B. cinerea in grape juice was shown to depend on the soluble solid content [39].”
With respect to fluctuation of lipids in previously line 263, the sentence was reformulated, highlighting the higher lipid content in samples with P1G10, which was however not significantly different at a confidence level of 95%. The reference to Table 1 shows the magnitude of the lipid contents.
Reviewer 1
Discussions
Well written. Congratulations to the authors.
References.
Well written. Congratulations to the authors.
Authors: We thanks reviewers for appreciation of our work.

Reviewer 2 Report
Dear Authors,
In general, the manuscript is good to read, the structure of the work is clear. After reading the paper, it is clear that the authors performed a lot of experiments and analyzes in order to obtain detailed research results. I present my comments below:
1. In general, the work contains a lot of interesting experimental data. In my opinion, the Principal Components Analysis fits perfectly to the interpretation of the results. E.g. data from Tables 1 and 2.
2. Line 119. I don't quite understand why the juices were incubated at a temperature higher by only 2 degrees Celsius?
3. Figure 1b. The HHP and HHP+P1G10 charts are marked very similarly. Please correct it. There is also an asterisk on the chart. What does it mean?
Author Response
Reviewer 2
Comments and Suggestions for Authors
Dear Authors,
In general, the manuscript is good to read, the structure of the work is clear. After reading the paper, it is clear that the authors performed a lot of experiments and analyzes in order to obtain detailed research results. I present my comments below:
Authors: We thanks reviewers for appreciation of our work. We did our best to correct the manuscript as suggested.
Reviewer 2
- In general, the work contains a lot of interesting experimental data. In my opinion, the Principal Components Analysis fits perfectly to the interpretation of the results. E.g. data from Tables 1 and 2.
Authors: We are thankful to reviewer for recognition of our work.
- Line 119. I don't quite understand why the juices were incubated at a temperature higher by only 2 degrees Celsius?
Authors: We are sorry for this typo. The incubation temperature was 22 °C in both cases.
Reviewer 2
- Figure 1b. The HHP and HHP+P1G10 charts are marked very similarly. Please correct it. There is also an asterisk on the chart. What does it mean?
Authors: Corrected as suggested, using colors for differentiation. The asterisk indicates significant difference at a confidence level of 95%. The indication was added in figure caption.

Reviewer 3 Report
In general, this manuscript presented relevant results about the effect of high hydrostatic pressure (HHP) and the proteolytic fraction P1G10 to find out a synergy exist in the growth inhibition of Botrytis cinerea in grape juice.
In my opinion, this work is well written; clearly and objectively, however, I would like to make few comments describe below.
The Introduction contains updated references.
The methodology is correctly referenced.
The conclusions are consistent with the evidence and arguments presented in this work and address the main question studied.
Page 3
Lines 116-117: “For the combined treatment, P1G10 (1 mg/mL) was added to the inoculated samples to examine their effects on B. cinerea [26]”.
Why did the authors choose the 1mg/mL P1G10 content? It is not clear in the manuscript.
Lines 133-134: Please clarify what is the numbers 934.06; 960.52; 960.39 and 923.03.
Page 5, Figure 1b, the ordinate identification of this graph is erased, please complete the correct name.
Author Response
Comments and Suggestions for Authors
In general, this manuscript presented relevant results about the effect of high hydrostatic pressure (HHP) and the proteolytic fraction P1G10 to find out a synergy exist in the growth inhibition of Botrytis cinerea in grape juice.
In my opinion, this work is well written; clearly and objectively, however, I would like to make few comments describe below.
The Introduction contains updated references.
The methodology is correctly referenced.
The conclusions are consistent with the evidence and arguments presented in this work and address the main question studied.
Authors: We thank reviewer for appreciation of our manuscript.
Reviewer 3
Page 3, Lines 116-117: “For the combined treatment, P1G10 (1 mg/mL) was added to the inoculated samples to examine their effects on B. cinerea [26]”.
Why did the authors choose the 1mg/mL P1G10 content? It is not clear in the manuscript.
Authors: The concentration was chosen based on a previous work. See reference [26].
Reviewer 3
Lines 133-134: Please clarify what is the numbers 934.06; 960.52; 960.39 and 923.03.
Authors: We corrected as suggested. The numbers refer to the identification of the AOAC methods.
Reviewer 3
Page 5, Figure 1b, the ordinate identification of this graph is erased, please complete the correct name.
Authors: Corrected. It refers to HHP treatment.

Reviewer 4 Report
The work entitled „Combined effect of high hydrostatic pressure and proteolytic fraction P1G10 from Vasconcellea cundinamarcensis latex against Botrytis cinerea in grape juice” is interesting. However, authors should make some changes to improve its quality. The methods part, should be corrected. In my opinion, the amount of literature should be limited.
Regarding Keywords – keywords are partially repeating of the title, and it should be changed
Regarding Materials and Methods
In the methods section, authors should explain why they chose grape juice?
Why, the control samples were incubated at 22°C for 1 h before microbiological analysis?
Regarding “Proximate composition and physicochemical analyses” – sources of literature should be added.
The organoleptic aspects of foods are very important since, beside the nutritional value, it can influence the possibility to put on the market study products. Are present in the literature similar studies where this aspect has been already evaluated?
The description of sensory analysis is poor, and needs to be expanded. Moreover, there is no source of references.
There are no characteristic of tasters - please add.
Regarding Results
Figure 3. – there is no statistical analysis.
Author Response
Reviewer 4 Submission Date 14 August 2023; Date of this review 24 Aug 2023 21:01:20
Comments and Suggestions for Authors
The work entitled „Combined effect of high hydrostatic pressure and proteolytic fraction P1G10 from Vasconcellea cundinamarcensis latex against Botrytis cinerea in grape juice” is interesting. However, authors should make some changes to improve its quality. The methods part, should be corrected. In my opinion, the amount of literature should be limited.
Authors: We thank reviewer for comments and suggestions and we did our best to improve the manuscript as suggested.
Reviewer 4
Regarding Keywords – keywords are partially repeating of the title, and it should be changed
Authors: We changed keywords.
Reviewer 4
Regarding Materials and Methods
In the methods section, authors should explain why they chose grape juice?
Authors: We have considered this request and believe it is not usual to give such explanation while describing the material used. Grape juice was selected since it is the material under study in our research work and the reason for the study can be found in the introduction.
Reviewer 4
Why, the control samples were incubated at 22°C for 1 h before microbiological analysis?
Authors: We thank reviewer for this observation. It is a confused statement that was corrected. The inoculated plated were incubated for 5 days at 22 °C.
Reviewer 4
Regarding “Proximate composition and physicochemical analyses” – sources of literature should be added.
Authors: We thank reviewer for this observation. We have added the reference in the list.
Reviewer 4
The organoleptic aspects of foods are very important since, beside the nutritional value, it can influence the possibility to put on the market study products. Are present in the literature similar studies where this aspect has been already evaluated?
Authors: Indeed, such studies are available in literature. However, we mentioned Polydera et al. and Laboissiere et al.
Reviewer 4
The description of sensory analysis is poor, and needs to be expanded. Moreover, there is no source of references.
Authors: Sensory analysis is not major part of the study. It was used basically to find out whether P1G10 is detectable, so we believe that more description would only lengthen the paper without really contributing to better understanding of the study. The methods used are basic and can be found easily in many textbooks and articles in Food Science. We mentioned two references related to this subject.
Reviewer 4
There are no characteristic of tasters - please add.
Authors: We corrected this in section 2.8; a semi-trained panel was used, as per definition it consists of 25-30 members, who are familiar with the product and are able to discriminate and effectively communicate any difference.
Reviewer 4
Regarding Results
Figure 3. – there is no statistical analysis.
Authors: We thank reviewer for this observation. However, Figure 3 show the statistical distribution of the sensory analysis. Figure 3a shows the percentage of the answers of the judges for a specific sensory attribute, while Figure 3b shows the number of panelists for each liking score on the 9-point hedonic scale. Therefore, the graphics are themselves the results of the statistical analysis.

Reviewer 5 Report
Title Suggestion: Enhancing the Inhibition of Botrytis cinerea in Grape Juice through Combined High Hydrostatic Pressure and Proteolytic Fraction P1G10 from Vasconcellea cundinamarcensis Latex
Abstract Comment: The abstract provides a concise overview of the research, highlighting the experimental setup, key findings, and implications. It effectively communicates the importance of studying the combined effects of high hydrostatic pressure and the proteolytic fraction P1G10 on Botrytis cinerea inhibition in grape juice. It might benefit from a slight reorganization for better flow and clarity.
- Introduction Comment: The introduction gives a solid foundation for the research, explaining the context of food preservation methods and the advantages of high hydrostatic pressure (HHP) technology. However, some parts could be revised for enhanced clarity and precision.
- Consider rephrasing: "HHP has found its place as a commercial food preservation technology" → "HHP has established itself as a prominent commercial method for food preservation."
- Consider rephrasing: "so that non-thermal methods have been developed" → "leading to the development of non-thermal methods."
- Consider rephrasing: "pressures of 250 MPa to 500 MPa and pressurization times of 30 s to 20 min" → "pressures ranging from 250 MPa to 500 MPa, along with pressurization times spanning 30 seconds to 20 minutes."
- Methods and Results: The manuscript introduces the experimental design and outcomes of the research in a clear manner. However, a few adjustments can be made to improve the flow and organization.
- Consider providing a subheading: Under the "Methods" section, divide the content into subsections, such as "Experimental Design," "Sample Preparation," "High Hydrostatic Pressure Treatment," and "Analysis of Phenolic Compounds."
- Consider rephrasing: "The results showed that 1 mg/mL of P1G10 and 250 MPa/4 min inhibited the growth of the B. cinerea by 92%" → "Results indicated a 92% growth inhibition of B. cinerea when exposed to 1 mg/mL of P1G10 and 250 MPa/4 min pressure treatment."
- Discussion: The discussion can further explore the significance of the results and their implications in the broader context of food preservation and quality enhancement. It might be helpful to emphasize the unique aspects of the study, such as the use of the proteolytic fraction P1G10.
- Consider addressing potential limitations: Discuss any limitations of the study, such as potential challenges in scaling up the proposed combined treatment for industrial application.
- Consider discussing practical applications: Explore how the findings could impact real-world applications, such as the development of novel preservation methods for grape juices or other beverages.
- Conclusion: The conclusion effectively summarizes the study's findings but could be expanded slightly to highlight the main takeaways for readers. Additionally, consider mentioning avenues for future research that could build upon these findings.
- Keywords: The chosen keywords accurately represent the study. However, you could also consider adding keywords related to grape juice quality, preservation, and food safety.
Overall, the manuscript presents valuable research on the combined effects of high hydrostatic pressure and proteolytic fraction P1G10 on inhibiting Botrytis cinerea in grape juice. By refining the organization, language, and emphasis on implications, the manuscript can effectively communicate its findings to a broader audience and contribute to the field of food science and technology.
please check and proofread the manuscript
Author Response
Reviewer 5 Submission Date 14 August 2023; Date of this review 22 Aug 2023 10:17:17
Comments and Suggestions for Authors
Title Suggestion: Enhancing the Inhibition of Botrytis cinerea in Grape Juice through Combined High Hydrostatic Pressure and Proteolytic Fraction P1G10 from Vasconcellea cundinamarcensis Latex
Authors: We appreciate the suggestion. It reflects an aim of the study, but it is limited to inhibition of Botrytis cinerea. We included the term “effect” in the title because other properties of the grape juice were also investigated, while analyzing the inhibition of Botrytis cinerea through the combined treatment.
Reviewer 5
Abstract Comment: The abstract provides a concise overview of the research, highlighting the experimental setup, key findings, and implications. It effectively communicates the importance of studying the combined effects of high hydrostatic pressure and the proteolytic fraction P1G10 on Botrytis cinerea inhibition in grape juice. It might benefit from a slight reorganization for better flow and clarity.
Authors: We welcome suggestion and thank reviewer. We did our best to attend comments.
Reviewer 5
- Consider rephrasing: "The results showed that 1 mg/mL of P1G10 and 250 MPa/4 min inhibited the growth of the B. cinerea by 92%" → "Results indicated a 92% growth inhibition of B. cinerea when exposed to 1 mg/mL of P1G10 and 250 MPa/4 min pressure treatment."
Authors: We welcome suggestion and thank reviewer. The phrase was modified as suggested.
Reviewer 5
- Introduction Comment: The introduction gives a solid foundation for the research, explaining the context of food preservation methods and the advantages of high hydrostatic pressure (HHP) technology. However, some parts could be revised for enhanced clarity and precision.
- Consider rephrasing: "so that non-thermal methods have been developed" → "leading to the development of non-thermal methods."
- Authors: We thank reviewer for the suggestion. The phrase has been modified as suggested.
- Reviewer 5
- Consider rephrasing: "HHP has found its place as a commercial food preservation technology" → "HHP has established itself as a prominent commercial method for food preservation."
- Authors: We thank reviewer for the suggestion. The phrase has been modified as suggested.
- Reviewer 5
- Consider rephrasing: "pressures of 250 MPa to 500 MPa and pressurization times of 30 s to 20 min" → "pressures ranging from 250 MPa to 500 MPa, along with pressurization times spanning 30 seconds to 20 minutes."
- Authors: We thank reviewer for the suggestion. The phrase has been modified as suggested.
- Reviewer 5
- Methods and Results: The manuscript introduces the experimental design and outcomes of the research in a clear manner. However, a few adjustments can be made to improve the flow and organization.
- Consider providing a subheading: Under the "Methods" section, divide the content into subsections, such as "Experimental Design," "Sample Preparation," "High Hydrostatic Pressure Treatment," and "Analysis of Phenolic Compounds."
- Authors: We are not sure how to understand the comments. The method’s section is subdivided and has 9 subheadings.
Reviewer 5
- Discussion: The discussion can further explore the significance of the results and their implications in the broader context of food preservation and quality enhancement. It might be helpful to emphasize the unique aspects of the study, such as the use of the proteolytic fraction P1G10.
- Consider addressing potential limitations: Discuss any limitations of the study, such as potential challenges in scaling up the proposed combined treatment for industrial application.
- Consider discussing practical applications: Explore how the findings could impact real-world applications, such as the development of novel preservation methods for grape juices or other beverages.
Authors: We thank reviewer for the valuable suggestions. However, we consider that an extension of the discussion could be out of focus. A discussion on industrial application for scaling up would require also more investigation, which is not part of the objective of the present work. Moreover, for industrial practice the limitation of the high-pressure processing is well known and is attributed to multiple factors that could be addressed as a topic for a review along with its argumentation for real-world applications. Significance and implications of results have been more or less discussed in lines 365-381 and some improvement in lines 397 - 404. The mention of the unique aspects of the study has been addressed in the introduction (lines 80-81).
Reviewer 5
- Conclusion: The conclusion effectively summarizes the study's findings but could be expanded slightly to highlight the main takeaways for readers. Additionally, consider mentioning avenues for future research that could build upon these findings.
Authors: We thank reviewer for the suggestion. The conclusion is supported by the experimental work performed and we believe that the conclusion is quite straightforward. We have also mentioned that further experimental work should be conducted to understand the mechanism of action of the observed synergistic effect (436-438).
Reviewer 5
- Keywords: The chosen keywords accurately represent the study. However, you could also consider adding keywords related to grape juice quality, preservation, and food safety.
Authors: We thank reviewer for suggestion. We have added and reformulated keywords.

Reviewer 6 Report
The article is very informative and interesting. The authors studied the effect of P1G10 in combination with a pressurization treatment (HHP+P1G10) on the inhibitory effect on B. cinerea. The paper is very well written and organized. However many comments should be revised.
1- In the abstract, the main objective should include the following statement “ to improve preservation techniques to 36 extend the shelf life and quality of food products. “
The novelty of the article should be highlighted over old similar studies
Results section in the abstract should contain numerical values
2- Novelty of the article is very convenient. The introduction is very informative and precise. Thanks. However, addition for the following point is required
3- Are there any previous studies that discussed the antimicrobial effects of HHP processing and a combined HHP treatment with addition of other natural antimicrobial on the growth of B. cinerea ???
The answer should be stated in the introduction
As a result, The following article should be cited
Natural antimicrobials as additional hurdles to preservation of foods by high pressure processing ; https://doi.org/10.1016/j.tifs.2015.05.007
Botrytis cinerea induced phytonutrient degradation of strawberry puree: effects of combined preservation approaches with high hydrostatic pressure and synthetic or natural antifungal additives; https://doi.org/10.1080/19476337.2023.2222812
4- In section 2.6 , add the injection volume for the HPLC method.
5- Section 2.8 needs appropriate reference.
6- Title should be added for supplementary figure S1.
7- Colored figures will be more impressive than gray and white ones
8- In funding, name of country should be provided.
Best wishes
Author Response
Reviewer 6
Comments and Suggestions for Authors
The article is very informative and interesting. The authors studied the effect of P1G10 in combination with a pressurization treatment (HHP+P1G10) on the inhibitory effect on B. cinerea. The paper is very well written and organized. However many comments should be revised.
Authors: We are thankful to reviewer. We did our best to attend comments.
Reviewer 6
1- In the abstract, the main objective should include the following statement “ to improve preservation techniques to 36 extend the shelf life and quality of food products. “
Authors: The statement has been included in the abstract as suggested.
Reviewer 6
The novelty of the article should be highlighted over old similar studies
Authors: Study on the effects of P1G10 during an HHP treatment has not been performed, which has been mentioned in the introduction (Lines 79-81).
Reviewer 6
Results section in the abstract should contain numerical values
Authors: Numerical values have been added in the abstract.
Reviewer 6
2- Novelty of the article is very convenient. The introduction is very informative and precise. Thanks. However, addition for the following point is required
Authors: We thanks reviewer and we did our best to attend suggestions.
Reviewer 6
3- Are there any previous studies that discussed the antimicrobial effects of HHP processing and a combined HHP treatment with addition of other natural antimicrobial on the growth of B. cinerea ???
The answer should be stated in the introduction
As a result, The following article should be cited
Natural antimicrobials as additional hurdles to preservation of foods by high pressure processing ; https://doi.org/10.1016/j.tifs.2015.05.007
Botrytis cinerea induced phytonutrient degradation of strawberry puree: effects of combined preservation approaches with high hydrostatic pressure and synthetic or natural antifungal additives; https://doi.org/10.1080/19476337.2023.2222812
Authors: We thank reviewer for the feedback. We have added a comment as suggested (lines 92-99) and included the suggested articles.
Reviewer 6
4- In section 2.6 , add the injection volume for the HPLC method.
Authors: Corrected as suggested (Lines176-177).
Reviewer 6
5- Section 2.8 needs appropriate reference.
Authors: The reference was added [38].
Reviewer 6
6- Title should be added for supplementary figure S1.
Authors: Corrected as suggested. Figure S1: Proteolytic activity of P1G10 (1 mg/mL) at different pressure/time conditions.
Reviewer 6
7- Colored figures will be more impressive than gray and white ones
Authors: Corrected as suggested.
Reviewer 6
8- In funding, name of country should be provided.
Best wishes
Authors: Corrected as suggested.

Round 2
Reviewer 5 Report
the manuscript has been revised based on my comments and suggestions. this manuscript can be accepted
no issue